# Ultrafast Fluorescence Spectroscopy via Upconversion and Its Applications in Biophysics

**DOI:** 10.3390/molecules26010211

**Published:** 2021-01-03

**Authors:** Simin Cao, Haoyang Li, Zenan Zhao, Sanjun Zhang, Jinquan Chen, Jianhua Xu, Jay R. Knutson, Ludwig Brand

**Affiliations:** 1State Key Laboratory of Precision Spectroscopy, East China Normal University, Shanghai 200062, China; simin_cao@163.com (S.C.); lee1205@vip.163.com (H.L.); 13464210900@163.com (Z.Z.); sjzhang@phy.ecnu.edu.cn (S.Z.); jqchen@lps.ecnu.edu.cn (J.C.); 2Laboratory for Advanced Microscopy and Biophotonics, National Heart, Lung and Blood Institute, National Institutes of Health, Bethesda, MD 20892, USA; 3Department of Biology, Johns Hopkins University, 3400 North Charles Street, Baltimore, MD 21218, USA; Ludwig.Brand@jhu.edu

**Keywords:** tryptophan, NADH, fluorescent RNA aptamer, upconversion, quasi-static self-quenching, solvation dynamics

## Abstract

In this review, the experimental set-up and functional characteristics of single-wavelength and broad-band femtosecond upconversion spectrophotofluorometers developed in our laboratory are described. We discuss applications of this technique to biophysical problems, such as ultrafast fluorescence quenching and solvation dynamics of tryptophan, peptides, proteins, reduced nicotinamide adenine dinucleotide (NADH), and nucleic acids. In the tryptophan dynamics field, especially for proteins, two types of solvation dynamics on different time scales have been well explored: ~1 ps for bulk water, and tens of picoseconds for “biological water”, a term that combines effects of water and macromolecule dynamics. In addition, some proteins also show quasi-static self-quenching (QSSQ) phenomena. Interestingly, in our more recent work, we also find that similar mixtures of quenching and solvation dynamics occur for the metabolic cofactor NADH. In this review, we add a brief overview of the emerging development of fluorescent RNA aptamers and their potential application to live cell imaging, while noting how ultrafast measurement may speed their optimization.

## 1. Introduction

The ability to monitor luminescence transients creates unique insights into the environment and dynamics of those luminescent molecules. In particular, biologically relevant fluorescence (from the allowed transitions of singlet states in extended organic molecules) is revealing about water dynamics, host macromolecule dynamics, and the detailed binding of fluorescent molecules to hosts [1,2,3].

In the last two decades, the time resolution of fluorescence has advanced strongly due to the popularization of the “upconversion” method, an optical method that breaks the ~40–100 ps (0.04–0.1 ns) resolution barrier set by direct photon timing in photomultiplier tube (PMT) detectors [4]. Instead of direct detection, fluorescence emission is directed through a nonlinear crystal that is “strobed” by a delayed (and usually infrared) laser pulse. This slices the emission into sub-ps (typically 300 fs = 0.3 ps = 0.0003 ns) packets that are “upconverted” when the energy of the strobe adds to those selected photons in the crystal. These packets are subsequently quantified by a slow photodetector. The transient is then traced by varying the delay of the strobe vs. excitation pulses, usually with a precision stage and retroreflector (to maintain constant overlap in the crystal).

The intrinsic fluorescent amino acid tryptophan (Trp) has, in particular, been revealing of dynamics in this way; both bulk water relaxation in and around protein (~2 ps) and slow protein-water coupled relaxation (~20 ps) have been discerned [5,6,7,8,9,10,11,12]. More importantly, a variety of ultrafast electron transfer (ET) processes from Trp to surrounding residues have been seen on the same 20–30 ps scale [13,14,15]. These processes lead to Quasi-Static Self-Quenching (QSSQ), a disparity between lifetime and yield due to “dark” fluorophores (e.g., those that ET process before their lifetime could be seen by conventional photodetector).

This well-developed QSSQ precedent in proteins led us to search out the role of QSSQ in other important biological fluorophores. One such example is the flexible dinucleotide NADH—the figurative “battery acid” of our cellular energy generators, mitochondria. From conventional photodetector studies, it was known that NADH free in solution exhibited short (~0.4 ns) decay times, but order of magnitude (2–8 ns) greater lifetimes were seen when it bound to cognate sites on target proteins. Our recent work showed the free molecule also exhibited a “dark” population, which, depending on detector, could hide 20–40% of the molecules from view [16,17]. Enzymatic binding abolished the quench.

We are moving on to another strong QSSQ candidate class, the group of folded RNA molecules that can host relatively nonfluorescent molecules in a planar pocket—a place that confers bright fluorescence—called “turn on aptamers” [18,19,20,21]. We explain why their lifetime/yield disparity implies dark subpopulations and suggest that ultrafast studies will soon guide their optimization.

## 2. Experimental Set-Up and Considerations

Upconversion is based on the pump–probe technique. Figure 1 shows one layout we developed especially for studying photophysics of biomolecules (e.g., Trp, NADH, and DNA). In brief, a seeded ultrafast Ti:sapphire regenerative amplifier (Astrella, Coherent Inc., Santa Clara, CA, USA) was used to generate a fundamental near-infrared pulse. The output amplified pulses centered at 800 nm typically have average power of 7 W and an autocorrelation pulse width of ~90 fs at a repetition rate of 1 kHz. The fundamental beam was routed to pump an optical parametric amplifier (OPerA Solo, Coherent, Inc., Santa Clara, CA, USA) to generate a mid UV excitation pulse (e.g., 340 nm) with an average power up to ~15 mW. The UV beam was spectrally cleaned and separated from the infrared beam and visible beam using a simple spectrometer comprised of a pair of UV prisms. Before being focused on the sample, the power of the excitation beam was carefully attenuated (typically to ~0.1 mW) to avoid any sample photodamage or photodegradation. The sample was held in a UV quartz disk cuvette with a diameter of 80 mm. The disk was spun continuously (tangential velocity >5 m/s) to avoid re/overexposure of the sample spot. The isotropic fluorescence was collected by a pair of off-axis parabolic mirrors and focused into a β-barium borate optical (β-${\mathrm{BaB}}_{2}O_{4}$, BBO) mixing crystal (usually 0.2 mm or 0.5 mm), together with a delayed infrared fundamental pulse (800 nm) as the “gate or strobe” pulse. By angle tuning the mixing crystal, the “upconverted” signal in the range of 230–280 nm was generated via type I sum frequency generation with the gate pulse, and that was directed to a monochromator (Omnik500, Zolix, Beijing, China) and conventional slow photomultiplier tube (CR317, Hamamatsu, Naka-ku Hamamatsu City, Japan).

A precise determination of the instrument response function (IRF) is essential in all ultrafast experiments, which can be determined by measuring the cross-correlation either between UV and the gate pulse or UV-generated Raman scattering of water and the gate pulse. In our system, both ways were used and the IRF was found to be around 350 fs (full width at half maximum, FWHM) as shown in Figure 2. In order to eliminate the influence of any E vector rotation contribution, magic-angle (54.7°) between excitation and collinear emission was used during the decay experiments. For experiments seeking rotation information, aka anisotropy measurement, the polarization of the excitation pulse was set to parallel (//) and perpendicular (⊥) to the crystal acceptance axis by a half-wave plate. The time-resolved anisotropy was then calculated by Equation (1):(1)$$r\left(t\right)=\frac{I_{//}-I_{⊥}}{I_{//}+2I_{⊥}}$$

Decay-associated spectra (DAS) and time-resolved evolution spectra (TRES) have been widely used for the study of ground-state heterogeneity, photochemical reaction, and solvation dynamics [13,14,15,16]. These spectra (with 10–15 nm resolution) are traditionally reconstructed by recording 10–20 kinetic traces I(t) at different wavelengths, which cover the whole emission region. A major drawback is that low spectrum resolution, together with long measurement time, limits the quality of obtained spectra. However, this drawback may be overcome with broadband upconversion spectrophotofluorometer concepts developed in the last two decades. Simultaneous measurement of the whole emission region of the fluorophore has been realized in various ways. The methods are: (i) rotation of the nonlinear optical (NLO) crystal, while synchronously scanning the detection monochromator [22]; (ii) wobbling the crystal while detecting with a spectrograph [23]; (iii) simultaneous phase matching across a broad spectral range followed by multiplex detection [24,25,26].

Figure 3 shows a scheme for fs-resolved broadband fluorescence upconversion spectrophotofluorometer developed in our laboratory, which is based on the third method. The fundamental pulse (800 nm) was directed into a parametric amplifier (TOPAS, Spectra-Physics) to generate the 1340 nm beam in horizontal polarization as the gate pulse. The output infrared beam was first widened and collimated by a telescope system, and then compressed by three prisms. The excitation pulse was provided by an optical parametric amplifier (OPerA Solo, Coherent Inc., Santa Clara, CA, USA). The excitation pulse was carefully attenuated and focused onto the sample cell, which was spun continuously (>5 m/s) to avoid overexposure. The fluorescence was collected by a concave mirror, and passed through a polarizer, allowing one to select vertical polarization. The vertically polarized fluorescence was refocused onto the potassium dihydrogen phosphate (KDP) crystal, together with the delayed gate pulse. The upconverted signal was generated through type II phase matching and separated from the unmixed fluorescence and gate harmonics by using a Glan polarizer. A spectrograph (Andor SR-500i-B1-R, Belfast, UK) combined with an EMCCD camera (Newton, Andor, Belfast, UK) was used to integrate and record the upconverted signal.

## 3. Solvation Dynamics of Biomolecules: Bulk Water and “Biological” Water

### 3.1. Solvent Relaxation of Tryptophan Alone in Water

Tryptophan has an extremely large Stokes shift, especially in polar solvent (i.e., water and physiological buffers). Most of the shift occurs only in the first few ps, which is undetectable even by the fastest PMT (~20 ps) [27]. Early studies by Ruggiero and coworkers [28] noted a ps transient with a lifetime of ~1.6 ps which was assigned to level crossing (internal conversion). Shen et al. [29] measured the time-resolved fluorescence spectra of Trp in water, from ~400 fs to 20 ps, with subpicosecond time resolution. Figure 4 shows the Trp emission at very early times measured at “magic angle” [29]. The fluorescence decay curves clearly show a fast decay at short emission wavelengths and a fast rise at longer emission wavelengths. Both the fast decay and the fast rise can be fitted with the same time constant (about 1.0–1.6 ps) with an average value of 1.2 ps. The 1.2 ps transient shows positive/negative amplitude at blue/red sides of the spectrum peak. Furthermore, the decay behavior was not dependent on the excitation wavelength and anisotropy measurement didn’t show a time constant matching those in fluorescence decay. The 1.2 ps component is clearly not from internal conversion (which would yield depolarization at the same time scale and amplitude that strongly depends on excitation wavelength). The most direct and reasonable explanation is solvent relaxation (bulk water).

Time-resolved fluorescence detected especially by frequency upconversion methods can provide much detailed experimental information about the solvation dynamics of biomolecules. Zewail and Zhong and coworkers [9,10] extended the solvation/hydration correlation function C(t) to the studies of solvation process in biomolecules (especially for tryptophan). The C(t) function has been defined by Equation (2):(2)$$C\left(t\right)=\frac{\nu_{s}\left(t\right)-\nu_{1}\left(t\right)}{\nu_{s}\left(0\right)-\nu_{1}\left(0\right)}$$
where $\nu_{s}\left(t\right)$ and $\nu_{1}\left(t\right)$ represent the function of overall emission maximum and apparent lifetime emission maximum with time, respectively. Figure 5 shows the typical hydration correlation function of tryptophan in bulk water [12]. The C(t) function can be fitted to a biexponential decay, with two lifetimes of 180 fs (20%) and 1.1 ps (80%), consistent with the theoretical analysis (the earlier <50 fs lifetime is not resolved).

### 3.2. Solvation Dynamics at the Surface of Proteins

Protein functions are closely related to the waters near its surface. The water at the surface of a protein forms a hydration layer that has been loosely termed “biological water”. The dynamics of biological water is quite different from those for bulk water, which has been demonstrated clearly by NMR, dielectric relaxation and solvation dynamics methods [12,30,31,32,33]. The unique characteristics of biological water in many proteins including monellin, melittin, staphylococcus nuclease, human serum albumin, apomyoglobin and others have been thoroughly studied by Zewail and Zhong et al. using single tryptophan as a hydration probe [7,8,10,11,12,34,35,36,37]. Figure 6 shows the X-ray ribbon structure and solvation correlation functions for wild type staphylococcus nuclease and four mutants (each with different neighboring charged residues) [37]. The solvation dynamics of these proteins shows significant difference compared with the results of tryptophan in bulk water. Particularly, the C(t) spectral shift functions show a (clearly slower) additional decay component with a response of tens of picoseconds. The correlation functions of all five proteins yield two components, a few picoseconds for bulk water and tens of picoseconds for “biological” water. No significant change among the decay models in the four mutants is seen, and this demonstrates that the longer lifetime component in the C(t) function is not sensitive to the charged side chains. As we will discuss below, C(t) may originate in either relaxation or quenching.

### 3.3. Solvation Dynamics in Free NADH

The ultrafast dynamics of reduced nicotinamide adenine dinucleotide (NADH) has been rarely studied. In our recent work, NADH spectral evolution was also found to show two distinct solvation dynamics terms on ps time scales [16]. Figure 7 gives the upconversion fluorescence decay transients at different emission wavelengths [17]. Similar to Trp, the fluorescence of NADH contains a fast decay at “bluer” emission wavelength and a fast rise at “redder” emission wavelengths in the earliest ~5 ps, which is typical for bulk water, reflecting the librational/rotational motions of water around the surface of NADH. Global fitting results in a positive/negative decay-associated amplitude spectrum component with a lifetime of 1.4 ps. NADH fluorescence exhibits an additional and slower component (~26 ps) with more positive/weakly negative amplitude, which in one view suggests there could be dissociable “biological water” hydration of NADH. Further, the anisotropy experiment resolved a rotational diffusion time of 141 ps. The volume of rotating unit calculated from the rotational diffusion time is ~599 ${Å}^{3}$, about 20% larger than the theoretically calculated results (493 ${Å}^{3}$), which is also expected from the hydration volume typical of soluble biomolecules.

## 4. QSSQ in Biomolecules

### 4.1. QSSQ of Tryptophan in Dipeptides and Proteins

Chen et al. [38] proposed the term quasi-static self-quenching (QSSQ) and predicted the presence of QSSQ in dipeptides from the “yield defects” compared with single tryptophan analogs. Further, Xu et al. [13] investigated the time-resolved fluorescence of tryptophan dipeptides of the form Trp-X and X-Trp, where X is another aminoacyl residue, using an ultraviolet upconversion spectrophotofluorometer with time resolution less than 350 fs. The fluorescence of a pH-independent tryptophan derivative *N*-acetyl-l-tryptophan-amide (NATA) was also measured for comparison. The set of fluorescence decay profiles were analyzed by global fitting techniques and the decay associated spectra (DAS) were constructed to help distinguish heterogeneity vs. solvent relaxation. Figure 8 shows the typical DAS of Trp-leu in water. Besides the familiar lifetime component of 1–2 ps for bulk water with “positive blue, negative red” signature in DAS, another rapid decay component (~23 ps) with positive amplitude at all measured wavelengths was also found in many of these dipeptides. Notably, the 23 ps DAS has similar shape and width to those found by time-correlated single-photon counting (TCSPC) apparatus. Solvent relaxation on that time scale should result in a positive/negative DAS, or at least a severely distorted (much narrowed) and blue-shifted DAS. As an aside, internal conversion between indole ^1^$L_{b}$ and ^1^$L_{a}$ is neglected since it has been found to happen faster than 50 fs [13]. Therefore, this 20–30 ps decay component must originate from a highly quenched conformer (likely rotameric) of Trp, which verified the previously predicted mechanism for QSSQ in dipeptides—the loss of quantum yield without significant change of mean lifetime (detected by conventional ns instrumentation).

Most importantly, this fast decay component was also found in proteins, such as monellin and γ-Crystallin [14,15]. Earlier studies by Chen et al. [39,40] have shown that the fluorescence of tryptophan in human γd-crystallin is highly quenched, which (by reducing residency in excited states) could protect the lens from any photoinduced damage. Further, the time-resolved fluorescence of several single-tryptophan mutants of crystallin was measured by Xu et al. [14], using both an upconversion spectrophotofluorometer on the time scale of 300 fs–100 ps and a time correlated single photon counting system on the time scale of 80 ps–10 ns. Figure 9 shows the properly normalized decay associated spectra of these single-tryptophan protein mutants. All of the DAS contain a 2 ps component with positive/negative amplitude signature, originating from the bulk water relaxation. Interestingly, another fast decay component in 50–60 ps region was also resolved. This component shows positive alpha even at redder emission wavelength, typical for QSSQ, demonstrating the existence of a highly quenched conformers within crystallin.

### 4.2. QSSQ in Free NADH

Since Chance and coworkers proved decades ago that the autofluorescence of NADH can reflect the cellular oxidation-reduction levels, NADH fluorescence has been widely used as a biomarker to track intracellular metabolic state by measuring either the autofluorescence intensity or lifetime [41,42,43,44]. Visser and coworkers systematically studied the ns-resolved fluorescence spectra of NADH [45]. They found the fluorescence decay of free NADH was biexponential, with two lifetimes of 250 ps and 690 ps. An equilibrium of folded and unfolded structures of NADH is usually assumed to explain the lifetime heterogeneity and intramolecular energy transfer phenomenon [46,47,48,49]. In recent years, however, some studies have suggested that the two lifetime components could originate from different configurations of the nicotinamide ring, rather than folded and extended conformers of the entire molecule [50].

Surprisingly, the ultrafast dynamics of NADH has been rarely studied. Boldridge et al. [51] reported an ultrafast component (~30 ps) with positive amplitude in all emission wavelengths (420–700 nm) by using a 2 ps resolution streak camera. Becker et al. [27] also saw an ultrafast component with a lifetime of ~50 ps in their newly developed ultrafast HPM detectors. More recently, time-resolved fluorescence of free NADH in solution has been investigated using our ultraviolet upconversion spectrophotofluorometer (IRF < 350 fs) [16]. Figure 10 shows the novel decay associated spectra of free NADH in solution [16]. The DAS for the two longer lifetimes (244 ps and 690 ps) has positive amplitude at all measured wavelengths, representing two different self-quenched conformation of the folded molecules (ground-state heterogeneity). Further, two solvation-like DAS with lifetime of 1.4 ps and 26 ps were also found. Amplitudes of both lifetimes show positive/negative characteristics. In one, the “zero crossing point” for 1.4 ps is located around 460 nm, near the emission peak of NADH, which is typical for pure bulk water relaxation. In the other, however, the “zero crossing point” for 26 ps is obviously located at longer wavelength (~480 nm). This crossing redshift could result from the mixing of solvent relaxation and QSSQ (quasi-static self-quenching). Pure solvent relaxation should result in “zero crossing point” near the spectral maximum unless a drastic loss of radiative rate somehow accompanies the energy loss (i.e., rate vs. wavelength drops much faster than Strickler–Berg equation predictions). The more likely presence of QSSQ with positive amplitude at similar timescales leads to a composite 20–30 ps DAS with redder zero crossing points [52]. The mostly positive DAS implies a “dark” population unseen in conventional fluorescence lifetime imaging microscopy (FLIM) setup.

## 5. Application of RNA “Turn-On” Aptamers in Live Cell Imaging

RNA has been recognized as an important player in cellular function and exhibits complex dynamics in cells. Since RNA lacks strong intrinsic fluorescence, it has been challenging to track and visualize RNA molecules in living cells [53]. To address this problem, RNA aptamers which can enhance the fluorescence of some specific small molecules up to several thousand times have been proposed and successfully applied in live cell imaging. Tsien and coworkers [19] developed the first RNA aptamer that can enhance the fluorescence of the triphenylmethane dye malachite green (MG) up to 2300-fold. However, MG dye shows significant cytotoxicity which limits its further application in live cell imaging. The pioneering work by Jaffrey et al. [20] reported a nontoxic RNA-fluorophore complex, termed Spinach, which emits a green fluorescence comparable in brightness with green fluorescent protein (GFP). It has been successfully tagged to 5S ribosomal RNA (rRNA), which was further visualized in living mammalian cells in real time, indicating the ability of Spinach to probe intracellular RNA dynamics in living cells.

Besides raw fluorescence enhancement, the chromophore binding affinity is another important factor to evaluate the aptamer’s potential in cell imaging. The relatively low binding affinity ($K_{D}=300-540\;nM$) of Spinach limits its application in single molecule RNA visualization. Dolgosheina and coworkers [21] reported the RNA Mango aptamer which binds thiazole orange dyes with nanomolar affinity ($K_{D}=3.2\;nM$), while enhancing fluorescence up to 1100-fold. The high affinity of Mango makes it possible to use the fluorophore as a “purification tag”. Using a design strategy of competitive ligand binding microfluidic selection, three new aptamers (Mango II, III, and IV) have emerged. Among them, Mango III shows the most improved fluorescent gain and affinity compared with the original Mango I aptamer [54]. Further, a structure-guided cognate mutant Mango-III (A10U), and a functionally reselected mutant iMango-III have been refined [55]. These Mango mutants show higher quantum yield (~0.66) and much improved fluorescence (~50% brighter than enhanced green fluorescent protein), making them valuable candidate tags for RNA visualization in living cells.

Detailed studies of the photophysical process of aptamer-fluorophore complexes should help us better exploit them as fluorescent tags and guide us to develop the new generations of RNA aptamer. There are few reports about the photophysical studies (especially time-resolved fluorescence) of aptamers. Figure 11 shows the comparison of photophysical properties of Mango-III and its several mutants [55]. Clearly, both Mango-III(A10U) and iMango-III exhibit higher quantum yields (0.66 and 0.64, respectively) compared to Mango-III (0.55). Since all four aptamer-TO1-Biotin complexes have similar extinction coefficient and emission peak, there shouldn’t be an obvious change for their radiative transition rates. In that case, quantum yield should be proportional to its fluorescence lifetime [56]. However, all the mutants show similar or even shorter lifetimes with respect to Mango-III. This unreasonable trend could be due to QSSQ contributions from ultrafast components (<100 ps) undetectable by TCSPC instrument [13,38]. In order to fully resolve any potential ultrafast quenching components, measurement of fs-resolved fluorescence by modern upconversion techniques is now called for.

## 6. Conclusions

The ultrafast upconversion of fluorescence is an optical sampling method with bandwidth of a few nanometers, whose time resolution is comparable to the hundred femtosecond pulses (10^−13^ s). This provides unprecedented insight into the formative stages of the fluorescent state, especially in biological macromolecules, and these earliest events include vibrational cooling, solvent (especially water) relaxation, and ultrafast charge transfer or Forster transfer. All of those kinetic processes reveal structural and dynamics information about the host macromolecule, information that is only peripherally probed by other (NMR, inelastic scatter) techniques. Recent years have seen an explosion of upconversion studies on proteins, but the next area of application will likely center upon nucleic acid structures, such as telomeres and aptamers, whose features are even more elusive and ephemeral.

## Figures and Tables

**Figure 1 molecules-26-00211-f001:**
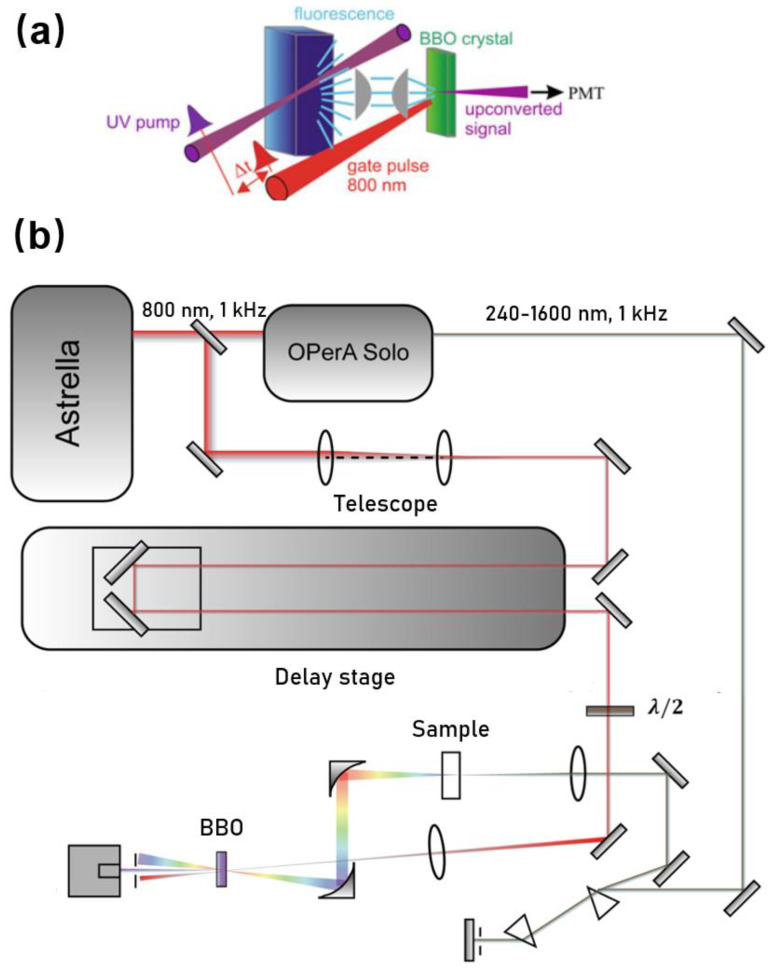
(**a**) Scheme of fs-resolved fluorescence upconversion and (**b**) the corresponding experimental setup.

**Figure 2 molecules-26-00211-f002:**
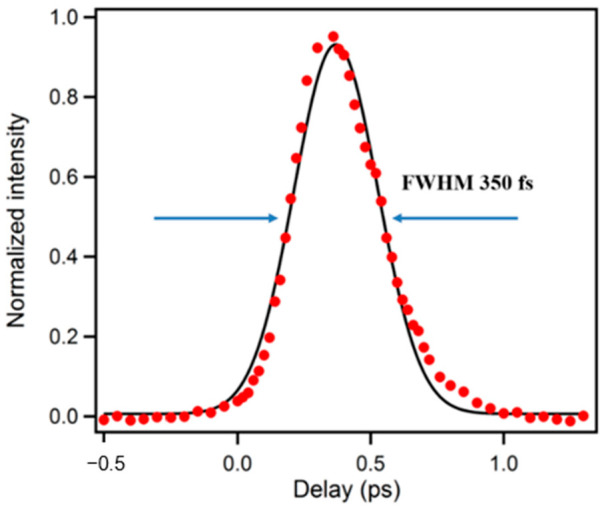
A typical pulse response function of upconversion spectrophotofluorometer.

**Figure 3 molecules-26-00211-f003:**
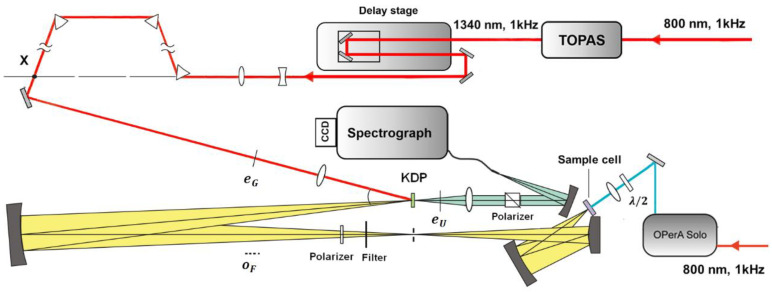
Scheme of fs-resolved broad band fluorescence upconversion setup.

**Figure 4 molecules-26-00211-f004:**
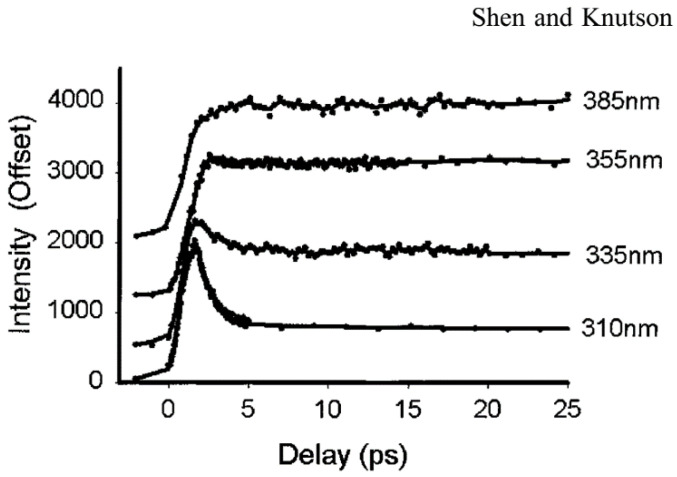
Upconverted fluorescence decay of tryptophan (Trp) at different wavelengths in water. Image reproduced from *Journal of Physical Chemistry B* 105 (2001) 6260.

**Figure 5 molecules-26-00211-f005:**
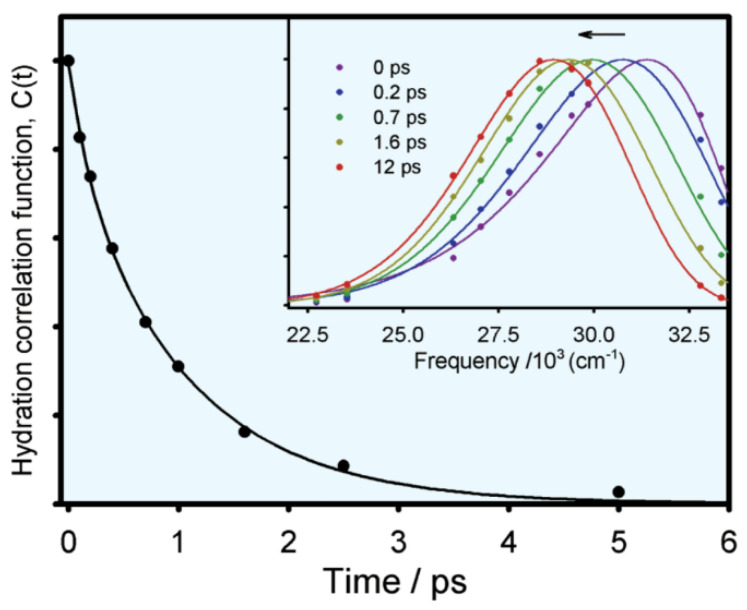
Solvent response function of tryptophan in aqueous solution (excitation: 288 nm). The insert is the normalized spectral evolution at five different decay points. Image reproduced from *Journal of Physical Chemistry B* 106 (2002) 12376.

**Figure 6 molecules-26-00211-f006:**
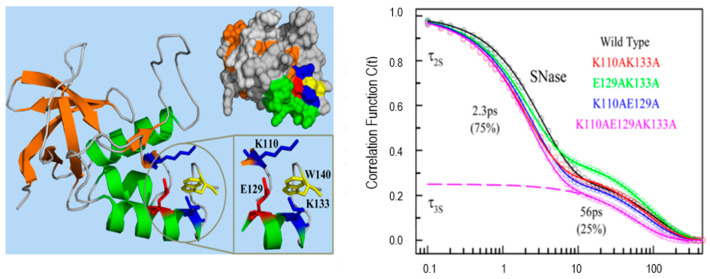
X-ray ribbon structure (left) and correlation functions (right) for wild type SNase and its four mutants. Three charged residues (K110, E129, and K133) are located around the single tryptophan W140. All proteins show two distinct relaxation times with a few and tens of picoseconds. Image reproduced from *Journal of Physical Chemistry Letters* 6 (2015) 5100.

**Figure 7 molecules-26-00211-f007:**
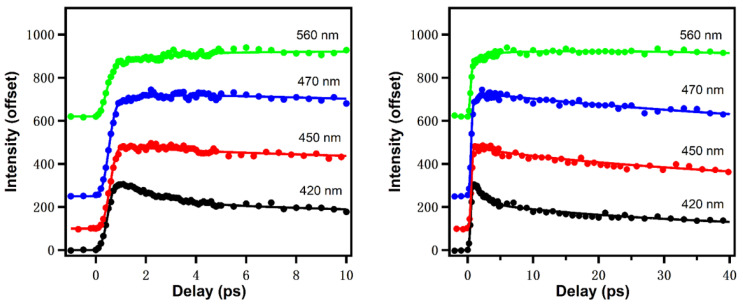
NADH emission at different wavelengths in different time windows (0–10 ps for left and 0–40 ps for right). Excitation wavelength: 340 nm. Image reproduced from *Journal of Physical Chemistry B* 124 (2020) 771.

**Figure 8 molecules-26-00211-f008:**
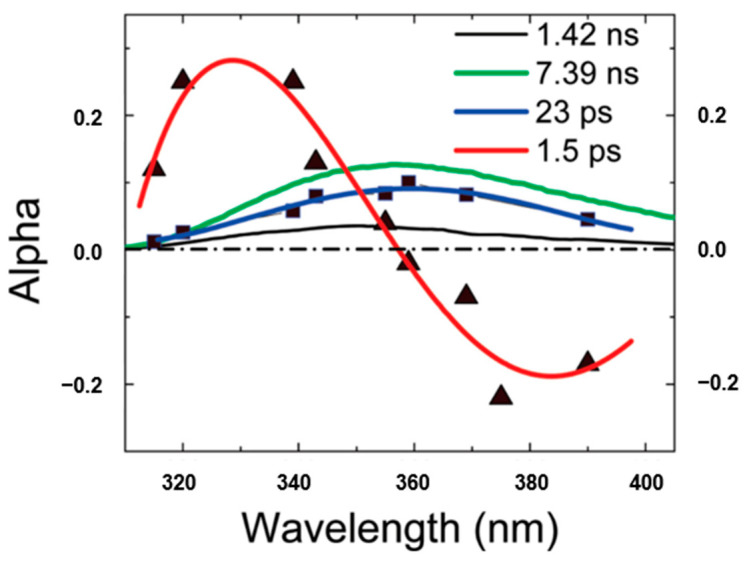
Decay associated spectra of Trp-leu in water. Image reproduced from *Journal of Physical Chemistry B* 113 (2009) 12084.

**Figure 9 molecules-26-00211-f009:**
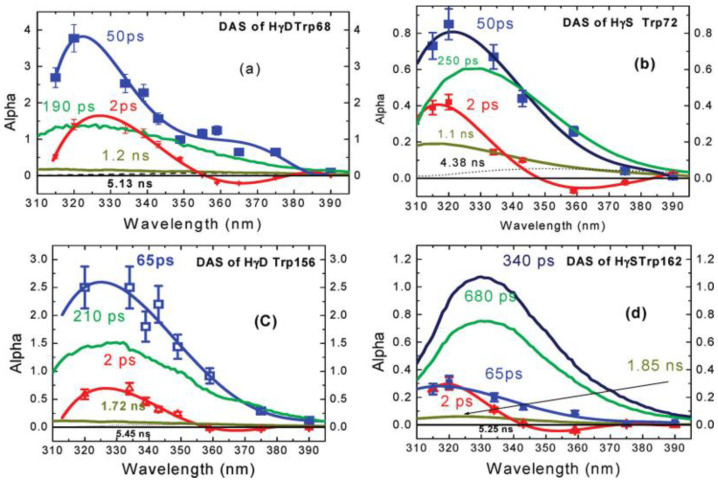
Decay associated spectra of tryptophan in four single-tryptophan mutants of crystallin. Image reproduced from *Journal of the American Chemical Society* 131 (2009) 16751.

**Figure 10 molecules-26-00211-f010:**
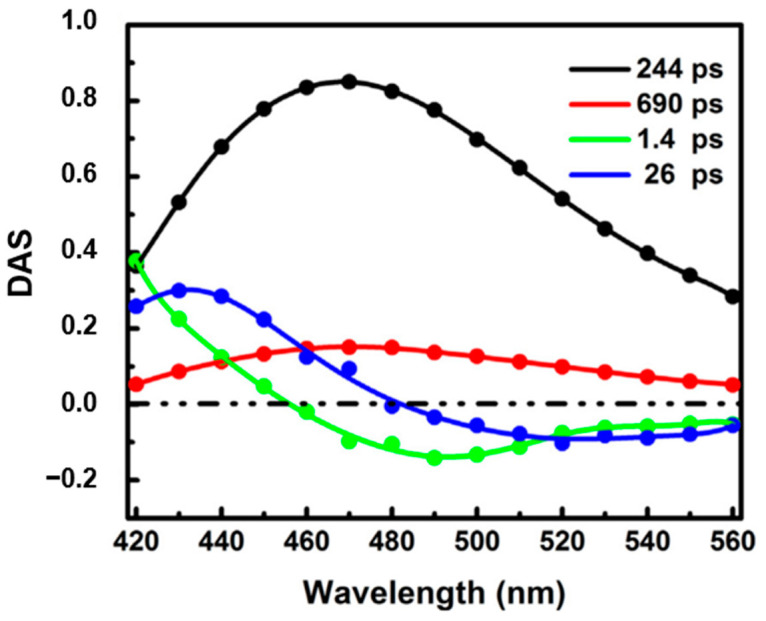
Decay associated spectra of free NADH in Tris-buffered solution (pH 7.35). Image reproduced from *Chemical Physics Letters* 726 (2019) 18.

**Figure 11 molecules-26-00211-f011:**
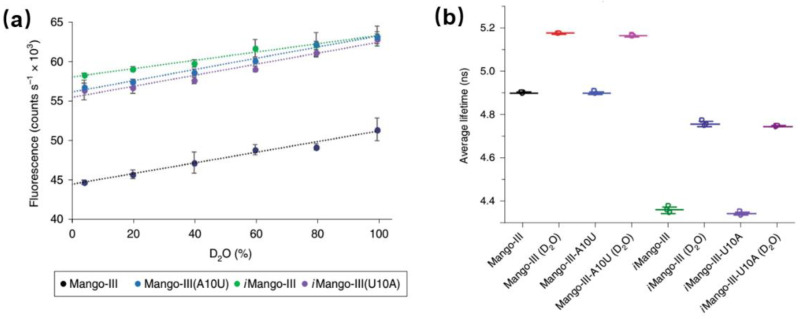
Photophysical properties of Mango-III and its several mutants. (**a**) Fluorescence intensity of four aptamer-TO1-Biotin complexes versus D_2_O percentage. (**b**) Fluorescence lifetime of four aptamer-TO1-Biotin complexes in pure water of 96.6% D_2_O. Image reproduced from *Nature Chemical Biology* 15 (2019) 472.

## Data Availability

Not applicable.

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
