# Peer review of "Ultrafast Fluorescence Spectroscopy via Upconversion and Its Applications in Biophysics"

_molecules, 2021, doi:10.3390/molecules26010211_

Round 1

Reviewer 1 Report

This review describes the method termed “upconversion”, in particular its application to ultrafast fluorescence studies in Biophysics.  In the past decade, upconversion methods have seen a rise in popularity due, in part, to applications to biological systems.  In fact, two of the author of this review (JRK and LB) have made fundamental contributions to this area.  This review focuses on the more important applications of upconversion to biological studies, including work on NADH, nucleic acids and intrinsic protein fluorescence.  They discuss in detail recent studies on NADH, which are particularly relevant given the increase in live cell studies on NADH as a monitor of metabolism.  The experimental aspects of upconversion are described in good detail which will allow anyone new to the field to appreciate what type of instrumentation, and hence cost, is required.  An excellent discussion of solvation dynamics of biomolecules is also provided.  This discussion will be especially appreciated by the tryptophan and protein fluorescence aficionados.  The discussion of quasi-static self-quenching (QSSQ) is also of interest, especially as it bears on recent interest in in vivo NADH fluorescence.  Finally, the overview of the emerging field of in vivo RNA aptamers is very valuable to readers – such as myself – who have considered entering this research area.  All in all I consider this review to be excellent, well-written and a valuable addition to the literature. 

Author Response

We thank the reviewer for all kind comments. All the english/spell changes have been marked with red color in the resubmitted manuscript.

Reviewer 2 Report

The manuscript presents nice work. This review paper deserves publication because it could be of interest to many readers.

However, the text looks to me somehow (just) like compilation of published works, it needs some polishing.

Many abbreviations are not explained, for example:

- TCSPC (page 8 and page 12) is not explained. NADH, NLO, BBO ...

Some typos:

page 3,

- line 85

ways were used and the IRF was found to be around 350 fs (FWHM) as shown in Figure X.

Figure 2?

- Fig 1 a) has only one scheme, "schemes" correspond to the whole Figure 1.

page 12,

line 300  "In order to fully resolved any potential ultrafast..." 

Some questions:

- line 86   how is the "magic-angle (54.7°)" determined?

- Figure 1: are BBO and BBO3 the same? Where are BBO1, BBO2...?

- First sentence of conclusion - something is missing?

"The ultrafast upconversion of fluorescence is an optical sampling method capable of time resolution comparable to the hundred femtosecond pulses (10-13s) the method typically employs, with bandwidth of a few nm."

Generally, the conclusion should be improved.

Author Response

The manuscript presents nice work. This review paper deserves publication because it could be of interest to many readers.

However, the text looks to me somehow (just) like compilation of published works, it needs some polishing.

1. Many abbreviations are not explained, for example:

- TCSPC (page 8 and page 12) is not explained. NADH, NLO, BBO ...

Author reply: We thank the reviewer for catching the misstatement, and we have added the full names of these abbreviations at the place where first mentioned.

2. Some typos:

page 3,

- line 85

ways were used and the IRF was found to be around 350 fs (FWHM) as shown in Figure X.

Figure 2?

- Fig 1 a) has only one scheme, "schemes" correspond to the whole Figure 1.

page 12,

line 300 "In order to fully resolved any potential ultrafast..." 

 Author reply: Thanks! The words “Figure X”, “Schemes” and “resolved” have been replaced with “Figure 2”, “Scheme” and “resolve”, respectively.

Some questions:

- line 86   how is the "magic-angle (54.7°)" determined?

Author reply: The contribution from anisotropy can be eliminated on magic-angle (54.7°) condition. The detailed derivation of magic-angle can be found in chapter 10 “Fluorescence Anisotropy” of the book “Principles of Fluorescence Spectroscopy, Third Edition”

- Figure 1: are BBO and BBO3 the same? Where are BBO1, BBO2...?

Author reply: Sorry for the mistake. Actually, there is only one BBO. We have reedited Figure 1 and the word “BBO3” in Figure 1 has been replaced with “BBO”.

- First sentence of conclusion - something is missing?

"The ultrafast upconversion of fluorescence is an optical sampling method capable of time resolution comparable to the hundred femtosecond pulses (10-13s) the method typically employs, with bandwidth of a few nm."

Generally, the conclusion should be improved.

Author reply: We are grateful for the favorable review and we have rephrased the sentences into “The ultrafast upconversion of fluorescence is an optical sampling method with bandwidth of a few nanometers, whose time resolution is comparable to the hundred femtosecond pulses ( s).” to make it clearer.

All the changes mentioned above have been marked with red color in the resubmitted manuscript.

Reviewer 3 Report

The present review article describes the experimental set up and applications of fluorescence upconversion. 

As for the present review, i can say that i enjoyed its reading and the message the authors want to convey with their paper appears clear to me, probably as a possible improvement would advice a more critical assessment of the systems presented. This possibly would help the reader to better address merits and shortcomings of the methodology.

In my opinion the article is clear and well written and can be published in its current form.

Author Response

We thank the reviewer for all kind comments. All the english/spell changes have been marked with red color in the resubmitted manuscript.

This manuscript is a resubmission of an earlier submission. The following is a list of the peer review reports and author responses from that submission.